# The Rate of Avoidable Pancreatic Resections at a High-Volume Center: An Internal Quality Control and Critical Review

**DOI:** 10.3390/jcm12041625

**Published:** 2023-02-17

**Authors:** Niccolò Surci, Christiane Sophie Rösch, Patrick Kirchweger, Lukas Havranek, Paul von Boetticher, Ines Fischer, Helwig Valentin Wundsam, Matthias Biebl, Reinhold Függer

**Affiliations:** 1Department of Surgery, Ordensklinikum Linz GmbH, Elisabethinen Hospital, 4020 Linz, Austria; 2Faculty of Medicine, Johannes Kepler University, 4020 Linz, Austria

**Keywords:** pancreaticsurgery, preoperative workup, clinical management, quality of care

## Abstract

Background: The incidence of benign diseases among pancreatic resections for suspected malignancy still represents a relevant issue in the surgical practice. This study aims to identify the preoperative pitfalls that led to unnecessary surgeries at a single Austrian center over a twenty-year period. Methods: Patients undergoing surgery for suspected pancreatic/periampullary malignancy between 2000 and 2019 at the Linz Elisabethinen Hospital were included. The rate of “mismatches” between clinical suspicion and histology was considered as primary outcome. All cases that, despite that, fulfilled the indication criteria for surgery were defined as minor mismatches (MIN-M). Conversely, the true avoidable surgeries were identified as major mismatches (MAJ-M). Results: Among the 320 included patients, 13 (4%) presented with benign lesions at definitive pathology. The rate of MAJ-M was 2.8% (*n* = 9), and the most frequent causes of misdiagnoses were autoimmune pancreatitis (*n* = 4) and intrapancreatic accessory spleen (*n* = 2). In all MAJ-M cases, various mistakes within the preoperative workup were detected: lack of multidisciplinary discussion (*n* = 7, 77.8%); inappropriate imaging (*n* = 4, 44.4%); lack of specific blood markers (*n* = 7, 77.8%). The morbidity and mortality rates for mismatches were 46.7% and 0. Conclusion: All avoidable surgeries were the result of an incomplete pre-operative workup. The adequate identification of the underlying pitfalls could lead to minimize and, potentially, overcome this phenomenon with a concrete optimization of the surgical-care process.

## 1. Introduction

Despite the steadily increasing improvement of all preoperative diagnostic tests—in particular the imaging techniques and the pathologic sampling tools—the incidence of benign diseases after pancreatic resections for presumed malignancies still represents a not negligible issue in the current surgical practice, ranging between 5% to 13% and showing no tendency to a significant decrease over time [1]. 

Although pancreatic neuroendocrine tumors (pNETs) and pancreatic ductal adenocarcinomas (PDACs) represent together up to 70% and thereby the great majority of the incidentally detected pancreatic solid lesions (PSLs), a wide range of non-neoplastic pathologies can mimic a solid malignancy, thus leading to a certain rate of misdiagnoses and, sometimes, to unnecessary resections [2,3]. In particular, chronic and autoimmune pancreatitis (CP, AIP) can occasionally appear at cross-sectional imaging as mass-forming inflammatory lesions, representing a considerable diagnostic challenge even for experienced radiologists [4]. Likewise, other rare tumour-like conditions—e.g., lymphoepithelial cysts, pancreatic hamartoma and intrapancreatic accessory spleen (IPAS)—are in some cases difficult to differentiate from PDAC and/or pNET only on the basis of imaging, markers and clinical presentation [5]. 

Although neoadjuvant therapy for locally advanced and borderline PDAC requiring pre-treatment histology is established, pancreatic resection without mandatory biopsy is the standard for primarily resectable pancreatic cancer. According to the current surgical practice and guidelines, the routine preoperative histologic proof of primary resectable PSLs—due to its potential risks and the unsatisfactory negative predicting value of most sampling techniques—is not recommended before performing surgical resections, unless the patients is unfit for major surgery or if other diagnoses need to be excluded [6,7,8]. Nevertheless, considering the constantly improving performances and reliability of endoscopic ultrasound (EUS)-guided tissue acquisitions and the potentially detrimental consequences of pancreatic surgery, preoperative pathology could still play a prominent role in resolving unclear and challenging clinical scenarios [2,9,10].

The main purpose of the present study is to analyze the rate of benign pathology among the resections for presumed malignancy at a single Austrian center in a timespan of twenty years, reviewing and discussing the eventual diagnostic mistakes and pitfalls occurred during the preoperative workup. 

## 2. Materials and Methods

This retrospective descriptive study analyzed all the patients who underwent an upfront pancreatic resection at the Department of Surgery of the Elisabethinen Hospital (Linz, Austria) between January 2000 and December 2019. The following inclusion criteria were defined (Figure 1): age ≥ 18 years; preoperative clinical and/or radiologic suspicion of pancreatic or periampullary solid malignancy, and pNET. Exclusion criteria included: patients with a conclusive preoperative histology; patients with suspicion (or cyto-/histological diagnosis) of intraductal papillary mucinous neoplasia (IPMN), mucinous cystic neoplasia (MCN), serous cystic neoplasia (SCN), CP, other rare malignancies or metastases involving the hepato-pancreato-biliary district (e.g., melanoma, renal cell carcinoma, sarcoma, etc.). Furthermore, those patients who underwent pancreatic resections after chemo- or radiotherapy (secondary resection) were excluded. 

All data were obtained from the prospectively maintained patient registry of the Department of Surgery. The pre- and postoperative clinical history of patients, as well as the most relevant demographic and clinical data were additionally checked and proofed through the review of the outpatient clinic reports, discharge letters, radiologic and operative reports. Additionally, in case of discrepancies between preoperative clinical suspicion and pathology report (see the section Outcomes, definition of “mismatch”) the clinical history and the available cross-sectional images (computer tomography and/or magnetic resonance) were independently reviewed from two surgeons (R.F. and N.S.) in order to identify eventual diagnostic mistakes and to assess the appropriateness of the whole diagnostic process. If required—i.e., when the reviewers’ judgements did not match each other—the cases were discussed with the other co-authors until a final shared assessment was reached.

This study was approved by the institutional Ethics Committee (Ordensklinikum Linz, EK 1225/2019) and was performed in compliance with the Good Clinical Practice standard and the principles of the Declaration of Helsinki.

### 2.1. Preoperative Workup

As previously reported by our research group [11], the preoperative diagnostic workup adopted at the considered center is depicted in Figure 2.

### 2.2. Outcomes

In order to define an accurate outcome parameter, the concept of “mismatch” was introduced. A mismatch indicates a substantial discrepancy between what was preoperatively expected—namely, pancreatic/periampullary malignancy or pNET—and the result of the definitive histological examination. A “minor mismatch” (MIN-M) represents the cases in which the pathology reported a different diagnosis that would have however met the indication criteria for surgery (e.g., diagnosis of duodenal adenoma with high-grade dysplasia after resection for suspected duodenal adenocarcinoma). The term “major mismatch” (MAJ-M) indicates those surgical resections that, based on the definitive histology, must be considered as not indicated or avoidable (e.g., diagnosis of autoimmune pancreatitis after resection for suspected PDAC).

Patients with histological proof of preoperatively suspected malignancy were classified as correctly predicted tumor (CPT). 

For every case of detected mismatch, a careful and comprehensive review of the preoperative medical investigations was performed with the aim to identify eventual mistakes, lacks or negligence in the diagnostic process. 

The incidence of MAJ-M and MIN-M was considered as primary outcome parameter. Secondary outcomes analyzed were morbidity and mortality in the mismatch and correctly predicted groups and the distribution of mismatches during the study period.

Complications were graded according to Clavien-Dindo classification [12]. Postoperative pancreatic fistula (POPF), postpancreatectomy hemorrhage (PPH) and delayed gastric emptying (DGE) followed the International Study Group (ISGPS) definitions [13,14,15].

### 2.3. Statistical Analysis

As a part of a retrospective internal “quality-control study”, the statistical analyses performed were primarily descriptive. No sample size calculation was performed due to the retrospective design of the study. Continuous variables are expressed as mean ± SD or as medians with interquartile range (IQR) as appropriate, whereas categorical variables are expressed as frequencies with percentages. Mismatches over resection periods were analyzed by the Welch’s (unequal variances) *t* test. A two-sided *p* value < 0.05 was considered as statistically significant. The statistical analysis was conducted using SPSS Statistic software version 26.0 (IBM Corporation, Armonk, NY, USA).

## 3. Results

Between 2000 and 2019, a total of 320 patients (Figure 1) underwent surgical resection for suspicion of periampullary or pancreatic malignancy. Most of the included patients were female (*n* = 171, 53.4%) with a mean age of 67.2 years (SD ± 10.2).

As reported in Table 1, the main clinical suspicion was represented by PDAC (*n* = 217, 67.8%), while 59 patients (18.5%) were addressed to surgery for a presumed periampullary malignancy (common bile duct, Vater’s papilla/ampulla or duodenum). In 44 (13.7%) patients the preoperative clinical diagnosis was pNET. (Table 1) 22 of 320 patients (6.9%) underwent an unsuccessful preoperative sampling attempt through EUS-guided fine needle aspiration (FNA) or fine needle biopsy (FNB); in all these cases the histological examination resulted not conclusive.

According to Table 2, the most frequently performed surgical procedure (61.9%) was pancreatoduodenectomy (pylorus preserving or Kausch-Whipple), followed by distal pancreatectomy (23.4%) and total pancreatectomy (14.7%). Thirteen (4.1%) of the resections were performed with minimally invasive technique (in all cases laparoscopic distal pancreatectomy). The median hospital stay was 14 days (IQR 11–20), and the overall rate of postoperative morbidity (Clavien-Dindo classification) was 41.2%. Patients who developed major complications (Grading ≥ III) amounted to 15.6% (*n* = 50). The rate of surgical reintervention was 8.1% (*n* = 26). The most frequent organ-specific complication was represented by POPF (*n* = 45, 14.1%), followed by DGE (*n* = 15, 4.7%), PPH (*n* = 11, 3.4%) and biliary fistula (*n* = 2, 0.6%). The 30-days postoperative mortality amounted to 2.5%. When considering only the mismatch cases (*n* = 13), the overall morbidity and mortality rates were 53.8% (7/13, median Clavien-Dindo Score = 2) and 0, respectively compared to 40.1% (123/307) and 2.6% (8/307) in the correctly predicted group.

The results of the definitive pathological reports are depicted in Table 3. In most of the cases (96.0%) the histological examination of surgical specimens confirmed the preoperative suspicion. In this regard, the most frequent diagnosis was represented by PDAC (68.8%), followed by pNET (11.6%).

In 13 patients (4%) a benign lesion was diagnosed. Overall, the number of MIN-M amounted to 4 (1.3%), while the detected MAJ-M added up to 9 (2.8 %). Among the MIN-M, the most frequent histological diagnosis was adenomyomatous hyperplasia of the Vater’s Papilla (*n* = 2), followed by adenomyoma of the common bile duct (*n* = 1) and pancreatic nesidioblastosis (*n* = 1). AIP was the most common cause of MAJ-M (1.3%, *n* = 4), followed by IPAS (0.6%, *n* = 2), lymphoepithelial cyst (0.3%, *n* = 1) and aspecific flogosis of the Vater’s papilla (0.3%, *n* = 1).In a further case of MAJ-M (0.3%, *n* = 1), no pancreatic lesions—except microscopic spots of pancreatic intraepithelial neoplasia (PanIN)—were found at the definitive pathology. 

The critical review of the preoperative workup revealed that the cause of misdiagnosis was in most of cases attributable to the incompleteness of the performed clinical, laboratory and/or radiological investigations (see Appendix A). Considering both MAJ-M and MIN-M, the most commonly identified pitfalls were the missing multidisciplinary discussion of cases (*n* = 10, 76.9%), the performance of an inappropriate or insufficient imaging (*n* = 7, 53.8%), and the lack of specific blood parameters—i.e., tumor markers or IgG4—measurements (*n* = 10, 76.9%).

A comparison of mismatches between the time periods 2000–2009, 2010–2014 and 2015–2019 is presented in Table 4. Statistical analysis revealed no differences in mismatches among the periods.

## 4. Discussion

This single-center experience describes the rate of benign histology at the definitive pathology after surgery for suspected malignancy in the last twenty years of surgical activity. Based on the assumption that the constant monitoring of surgical practice represents a key-point for the continuous improvement process in quality, this study aimed to identify and comprehensively review all the cases in which an unnecessary and potentially avoidable pancreatic resection was identified [16].

As reported in the literature, the incidence of benign disease among patients undergoing pancreatic resection for suspicion of malignancy is extremely variable, ranging from 5% to 35% of cases [3,17,18]. Considering these numbers, the 4% overall rate of misdiagnoses emerging from our retrospective analysis was encouragingly low. Furthermore, not all cases of reported mismatch underwent an absolutely inappropriate surgery. Indeed, 1.3% of misdiagnoses in our series was classified, based on the definitive pathology, as MIN-M; in other words—despite the baseline diagnostic mistake—the final pathology represented in these patients a concrete and suitable indication for surgery. For example, a pancreas nesidioblastosis causing severe episode of hypoglycemia and showing resistance to the medical therapy—as in the case #9, Appendix A—may require a partial or total pancreatectomy to achieve the complete resolution of symptoms [19]. Similarly, in most cases of adenomyoma of the Vaterian system (common bile duct, ampulla or papilla), surgical resection is often considered unavoidable, since imaging and endoscopy rarely offer a definitive diagnosis, allowing to exclude malignancy (Appendix A, #5, #10 and #11) [20,21]. 

The true rate of avoidable resection was represented by the 2.8% (*n* = 9) of cases which were classified as major mismatches (MAJ-M). As expected, autoimmune pancreatitis (AIP) was the most frequent cause of MAJ-M (1.3%, *n* = 4). Indeed, most of the lesions mimicking a PSL fall into the category of inflammatory and fibrotic conditions, like the “mass forming” CP, the groove/paraduodenal pancreatitis (GP/PDP), or the AIP [18]. This last entity, in particular, still represents a clinical challenge, since its diagnosis and treatment have not yet been unequivocally standardized [22]. Despite several efforts to establish reliable diagnostic criteria, many AIP cases are still missed or mistaken for PDAC [23]. Furthermore, diagnostic performance of EUS-guided tissue sampling methods in this field has been demonstrated to be poor, with an overall accuracy of 60% [24]. The real strength of endoscopic biopsy may actually be facilitating the IgG 4 immunostaining. In type I AIP IgG4 positive cells were founded in 85.9% of specimens compared with 3.4% in pancreatic adenocarcinoma [25]. In our series, one out of four patients underwent a preoperative tissue acquisition, and the only sampling attempt gave an inconclusive result. Beyond this, the most frequently detected pitfall among AIP patients (3/4 cases, 75.0%)—in line with the findings of other studies [26]—was the lack of the recommended preoperative serum measurements of Immunoglobulin G 4 (IgG4). However, IgG 4 sensitivity and specificity are known to vary considerably in studies and especially AIP type 2 is IgG4 negative in most cases [27]. According to The International Consensus Diagnostic Criteria for AIP [28], five main criteria are defined, namely histological findings, parenchymal and pancreatic duct imaging, other organ involvement, response to steroids and laboratory findings. Thus, normal IgG4 cannot rule out an AIP and diagnosis remains complex, especially differentiation from pancreatic adenocarcinoma. Today, pushing endoscopic biopsy and accurate weighing up of diagnostic criteria seem the only possibilities to further minimize the rate of inappropriate pancreatic resections in AIP patients. 

The second cause of detected MAJ-M in our series was represented by IPAS (0.6%, *n* = 2). IPASs are common findings in surgical specimens and their prevalence in the general population amounts approximately to 10% [29,30]. These lesions are frequently described at the cross-sectional imaging as hypervascular masses and can be radiologically indistinguishable from pNET; their most common location is the splenic hilum, even if in 10–15% of cases they are found in the pancreatic tail, posing a diagnostic predicament [31]. The commonly adopted diagnostic protocol foresees the performance of a tri-phasic abdominal CT, followed by a superparamagnetic iron oxide based (SPIO) magnetic resonance imaging (MRI) and/or a 99 m Technetium-labelled red blood cell (HD-RBC) scintigraphy; if these investigations result inconclusive or not diagnostic, an EUS-guided FNA or FNB is recommended [32]. In both our IPAS patients, the preoperative workup resulted insufficient, due to the lack of a complete set of imaging or to the missing attempt of tissue sampling. One of our patients had a negative red blood cell scintigraphy, underlining the diagnostic challenge.

In one case (#2) the final pathology did not report the presence of any solid pancreatic lesion. In principle, the spontaneous regression (SR) of PDAC is considered an extremely rare—or even doubtful—event [33]. Since in this case the presence of a hypervascular head pancreatic mass at the preoperative CT-scans was confirmed by the reviewers, this phenomenon could likely result from the total remission of a previously undiagnosed mass-forming flogosis (e.g., CP or AIP). 

Attempting an overview of the MAJ-M population, the most impressive and unexpected result we found is that 7 cases (77.8%) did not undergo a multidisciplinary preoperative discussion. The adoption of multidisciplinary team meetings (MDTMs)—proving to enhance the quality of care in terms of consistency, continuity, coordination, and cost-effectiveness, with a considerable positive impact on clinical outcomes—has become a cornerstone in the oncologic surgical practice in the last decades [34]. In particular, it has been demonstrated, that the interdisciplinary discussion of cases scheduled for surgery allows the achievement of three main goals: (1) better adherence to the referral guidelines; (2) optimization of treatment plans; (3) mitigation of the individual variability influencing the decision-making process [35,36]. Consequently, MDTMs lead to a definitively higher degree of standardization of the clinical and therapeutical decisions. Furthermore, as recently reported by the Verona SpaRo group, a further implementation of the level of care can be obtained through the integration of an additional monodisciplinary surgical meeting in the preoperative period, which facilitates the anticipation of pre- and intra-operative pitfalls and reduces potential single-observer mistakes [37]. In line with the globally shared recommendations in the field of pancreatic surgery, our Institution disposes of a multidisciplinary Tumor Board Conference, whose first documentations date back to 2003. Currently, all patients addressed to surgery at the Linz Elisabethinen Hospital routinely need the approval of the dedicated interdisciplinary expert team. The long study period may have caused bias by differences in diagnostic process and patient management procedures. However, we could not identify a significant divergence over the study period; indeed, avoidable pancreatic resections happened also recently. However, apart from the concrete motivations, missing preoperative MDTM should be considered the most relevant avoidable pitfall among the preoperative critical points identified in this analysis. 

Moreover, some other critical issues are worth to be mentioned. First of all, the three-phasic abdominal CT with pancreas protocol—which represents a gold standard in the radiological diagnosis of pancreatic masses—was not performed in 44.4% (4/9) of MAJ-M patients [38]. This maybe attributable to the early phase of our study period. Second, in cases #3 and #4 (suspected PDAC and carcinoma of the Papilla of Vater, respectively) the serum values of Ca 19.9 and CEA were not measured. Despite their limited specificity, these tumor markers are a fundamental part of the preoperative laboratory examination and may concretely help the clinicians making an accurate differential diagnosis between hepato-pancreato-biliary malignancies [39]. Finally, only 4 patients (44.4%) within the MAJ-M group underwent a preoperative tissue sampling (two attempts of FNA and two of FNB). Interestingly, in all cases the pathology report failed to reach diagnosis, resulting as not conclusive. Although a comprehensive evaluation of the role played by the preoperative histological typing—as well as its impact on the frequency of unnecessary resections—goes beyond the aim of this study, some relevant considerations should be addressed. As well known, the preoperative tissue sampling does not always allow to unequivocally solve the demanding issue of misdiagnosis [40]. In case of resectable PSL, the current tendency at most of the high volume pancreatic surgical centers—supported by the guidelines and experiences reported in the literature [6,7]—is to directly schedule the patients for operation without a previous tissue acquisition, unless (i) a metastatic or borderline/locally advanced tumor or (ii) a diagnosis of AIP are suspected [41,42]. This kind of policy—which has not changed during the past two decades [43]—is primarily justified by the limited and still not completely defined accuracy and negative predictive value (NPV) of CT-, US- or EUS-guided sampling techniques, as well as by their possible complications, e.g., post-procedural acute pancreatitis and tumor seeding [11]. Despite this, some authors documented the better performances—e.g., higher specificity and sensibility—obtained from the adoption of FNB-sampling—thanks also to the introduction of new specifically designed needles—especially when associated with the rapid on-site evaluation (ROSE) of the specimens [44]. Furthermore, the number of published trials evaluating the potential benefits of neoadjuvant chemotherapy (NAT) also for primary resectable PDACs—a scenario that would mandatory require a preoperative histological diagnosis - is constantly growing [45,46]. Despite our findings appear to discredit the concrete usefulness of the preoperative sampling, we strongly believe—in line with other authors [2]—that the above-mentioned aspects, together with the not negligible rate of morbidity and mortality affecting pancreatic resections, should lead to reconsider the role of cytology/biopsy within the diagnostic process. Unfortunately, the paucity of available data did not allow us to nearly investigate this topic.

This study has several limitations, first of all its retrospective nature. Despite the efforts made to enhance the reliability of the clinical data—e.g., accurate revision of the available paper and digital reports, imaging review and cross-matching of different internal databases—a partial lack of information due to the long study period could not be avoided. For the same reason—namely, the partial incompleteness of data, especially in the predigital era—the real impact of preoperative tissue sampling on the surgical outcomes—i.e., the rate of unnecessary resections—could not be objectively estimated; to achieve this goal, the conduction of further preferably prospective and multicentric studies is recommended. Finally, the critical revision of the preoperative indications and workup—even if founded on evidence-based and broadly shared clinical guidelines—was to some degree influenced by the examinator’s subjective surgical experience and beliefs. This bias may have been partially mitigated by the above-mentioned independent reviewing process performed in parallel by two or more of the authors. 

## 5. Conclusions

The rate of benign histological diagnoses detected after pancreatic resection for suspected SPL at a single high-volume center over the last twenty years was in line—or even lower—with what is reported in the current literature. In our experience, all avoidable pancreatic resections detected were the result of modifiable causes—like inaccuracy, oversight, negligence, or simple avoidable mistakes—occurred during the preoperative workup. The adequate identification of these failures and weaknesses could represent a key point to obtain a concrete improvement and optimization of the whole surgical-care process. For this reason, attaining a 0% ideal rate of unnecessary resections—even if challenging—should be considered an achievable goal rather than an utopia.

## Figures and Tables

**Figure 1 jcm-12-01625-f001:**
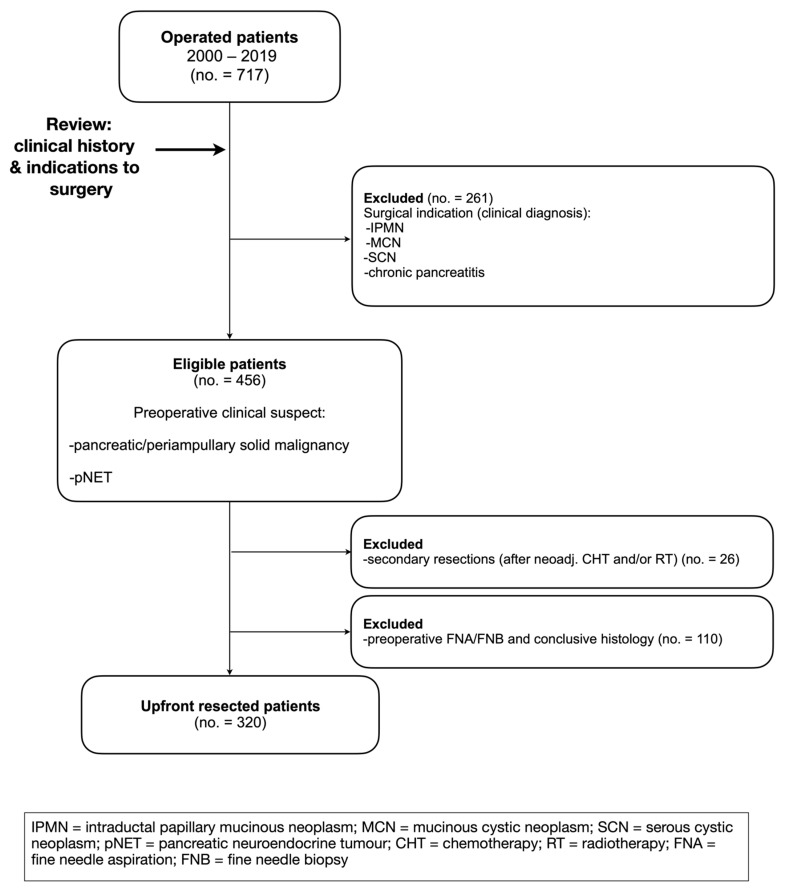
Inclusion and exclusion criteria.

**Figure 2 jcm-12-01625-f002:**
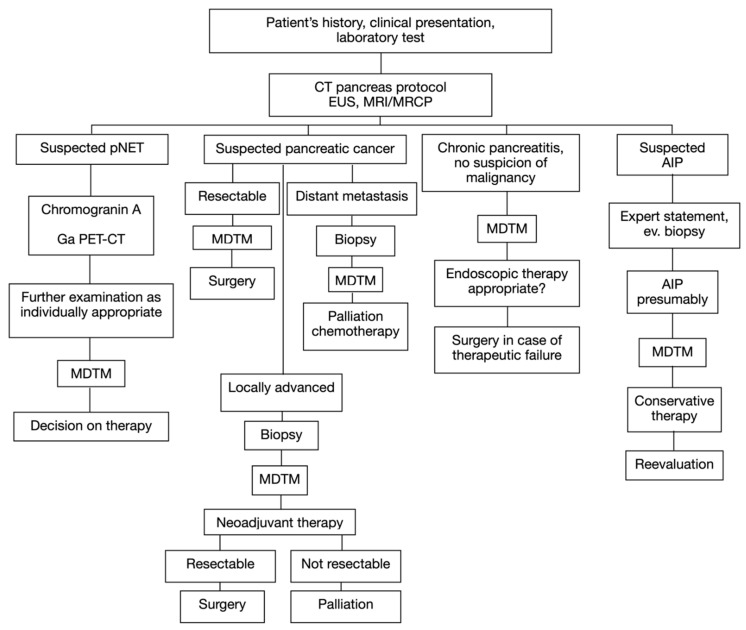
Diagnostic algorithm of a patient with a solid pancreatic mass. EUS = endoscopic ultrasound; MRI = magnetic resonance imaging; MRCP = magnetic resonance cholangiopancreatography; NET = neuroendocrine tumor; AIP = autoimmune pancreatitis; Ga PET-CT = Gallium-positron emission computed tomography; MDTM = multidisciplinary team meeting.

**Table 1 jcm-12-01625-t001:** General features and clinical suspicion (*n* = 320).

**Sex (no., %)**	
M	149 (46.6%)
F	171 (53.4%)
**Age (years, mean ± SD)**	67.2 (±10.2)
**Preoperative clinical suspicion (no., %)**	
PDAC	217 (67.8%)
Periampullary malignancy	59 (18.5%)
pNET	44 (13.7%)

SD = standard deviation; PDAC = pancreatic ductal adenocarcinoma; pNET = pancreatic neuroendocrine tumor.

**Table 2 jcm-12-01625-t002:** Surgery (*n* = 320).

**Type of surgery (no., %)**	
pancreatoduodenectomy (—PP or Kausch-Whipple)	198 (61.9%)
distal pancreatectomy	75 (23.4%)
total pancreatectomy	47 (14.7%)
**Complications (Clavien-Dindo Grading System) (no., %)**	
0	188 (58.8%)
I	32 (10%)
II	50 (15.6%)
IIIa	24 (7.5%)
IIIb	10 (3.1%)
IVa	6 (1.9%)
IVb	2 (0.6%)
V	8 (2.5%)
**Organ-specific complications (no., %)**	
POPF	45 (14.1%)
DGE	15 (4.7%)
PPH	11 (3.4%)
Biliary fistula	2 (0.6%)
**Reintervention**	26 (8.1%)
**Hospital stay (median, IQR, days)**	14 (11–20)
**30-days mortality (no., %)**	8 (2.5%)

PP = pylorus preserving; POPF = postoperative pancreatic fistula; DGE = delayed gastric emptying; PPH = post-pancreatectomy hemorrhage; IQR = interquartile range.

**Table 3 jcm-12-01625-t003:** Pathology (*n* = 320).

**Pathology report (no., %)** **Malignant or pNET** PDAC Papilla/Ampulla Ca. Choledochus Ca. Duodenal Ca. pNET **Benign** no lesion, Pan-In (MAJ-M) accessory spleen (MAJ-M) autoimmune pancreatitis (MAJ-M) adenomyoma choledochus (MIN-M) flogistic Vater’s papilla (MAJ-M) adenomyomatous hyperplasia Vater’s papilla (MIN-M) lymphoepithelial cyst (MAJ-M) nesidioblastosis (MIN-M)	220 (68.8%) 22 (6.9%) 18 (5.6%) 10 (3.1%) 37 (11.6%) 1 (0.3%) 2 (0.6%) 4 (1.3%) 1 (0.3%) 1 (0.3%) 2 (0.6%) 1 (0.3%) 1 (0.3%)

pNET = pancreatic neuroendocrine tumor; PDAC = pancreatic ductal adenocarcinoma; Ca. = Carcinoma; Pan-In = pancreatic intraepithelial neoplasia.

**Table 4 jcm-12-01625-t004:** Mismatches over resection periods.

	2000–2009	2010–2014	2015–2019
**Total mismatches**	2 (1.9%) ^A^	7 (7.1%) ^B^	4 (3.4%) ^C^
**MIN-M**	0 (0%)	2 (2.0%)	2 (1.7%)
**MAJ-M**	2 (1.9%)	5 (5.1%)	2 (1.7%)
**Total resections**	101	99	120

*p*-values (Welch’s *t*-test): A vs. B = 0.085; B vs. C = 0.224; A vs. C = 0.531.

## Data Availability

The data that support the findings of this study are not publicly available because they contain information that could compromise the privacy of research participants, but are available upon request from R.F.

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
