# Peer review of "The Rate of Avoidable Pancreatic Resections at a High-Volume Center: An Internal Quality Control and Critical Review"

_jcm, 2023, doi:10.3390/jcm12041625_

Round 1
Reviewer 1 Report
Surci et al. present an interesting review of their pancreatic resections of the last 20 years with focus on identifying unnecessary resections and the causes of insufficient indication. The paper is well written und of interest. The discussion and conclusion is justified.
However, I have some concerns:
1. The main concern is the choice of the inclusion criteria:
- There were included 110 patients with preoperative biopsy and conclusive histology (table 1). These patients must be excluded, as the definitive histology is already existent. Or are there patients with differences of preoperative histology and pathological histology of surgical specimen? If yes, how many and which mismatch?
- Likewise, pNETs that have already been histologically verified (see inclusion criteria) should be excluded, since the diagnosis is already known.
2. The rate of total pancreatectomies is quite high (21%). What is the reason for that?
3. Major complications is defined as grade ≥ III. That should be corrected in the results sections (page 5, line 146).
Reviewer 2 Report
I read with great interest this study, which aimed to add a piece of knowledge about the mismatches between clinical suspicion and final pathology leading to avoidable pancreatic resection. Specifically, the Authors performed a retrospective descriptive analysis on the patients who underwent pancreatic resection for presumed malignancy in a single institution. The outcomes of interest were the discrepancy between the preoperatively hypothesized malignant behavior of the lesion and the result of the definitive pathological examination, as well as morbidity and mortality rates in mismatched vs correctly hypothesized patients.
Some comments about the study:
Introduction:
1. Line 50: delete “absolutely”.
2. Line 64: delete “either sex”.
Discussion
1. Line 191: replace “incidence” with “rate”.
Tables
1. Table 1: the alignment of the bottom of the table ("Preoperative histology") is unclear.
Comment
Interesting paper on a current topic. The described rate of mismatches and the hypothesized causes that can lead to them are remarkable. The generalizability of the presented results is limited by their purely descriptive nature and the relatively limited number of events of interest, but they shed light on a topic that deserves to be explored in order to improve the pre-operative work-up.
However, from my point of view, patients with pre-operative FNA or FNB positive for cancer should not be included in the analyses. In fact, given the high positive predictive value of this assessment, these cases could present lower rates of mismatch, inflating the correctly diagnosed group and lowering the rates of mismatch.
It should be specified which patients had a positive pre-operative biopsy and separate analyses only in patients who did not undergo biopsy should be performed.
Round 2
Reviewer 1 Report
Thank you for your improvements!
Best regards!
Reviewer 2 Report
The manuscript has been revised accordingly to the comments.